# Multi-Omics Approach Reveals Dysregulation of Protein Phosphorylation Correlated with Lipid Metabolism in Mouse Non-Alcoholic Fatty Liver

**DOI:** 10.3390/cells11071172

**Published:** 2022-03-30

**Authors:** Sora Q. Kim, Rodrigo Mohallem, Jackeline Franco, Kimberly K. Buhman, Kee-Hong Kim, Uma K. Aryal

**Affiliations:** 1Department of Nutrition Science, Purdue University, West Lafayette, IN 47907, USA; kim2713@purdue.edu (S.Q.K.); kbuhman@purdue.edu (K.K.B.); 2Bindley Bioscience Center, Purdue Proteomics Facility, Purdue University, West Lafayette, IN 47907, USA; ferreir@purdue.edu (R.M.); francoj@purdue.edu (J.F.); 3Department of Comparative Pathobiology, Purdue University, West Lafayette, IN 47907, USA; 4Department of Food Science, Purdue University, West Lafayette, IN 47907, USA; keehong@purdue.edu

**Keywords:** NAFLD, fatty liver, HFD, mass spectrometry, proteomics, lipidomics

## Abstract

Obesity caused by overnutrition is a major risk factor for non-alcoholic fatty liver disease (NAFLD). Several lipid intermediates such as fatty acids, glycerophospholipids and sphingolipids are implicated in NAFLD, but detailed characterization of lipids and their functional links to proteome and phosphoproteome remain to be elucidated. To characterize this complex molecular relationship, we used a multi-omics approach by conducting comparative proteomic, phoshoproteomic and lipidomic analyses of high fat (HFD) and low fat (LFD) diet fed mice livers. We quantified 2447 proteins and 1339 phosphoproteins containing 1650 class I phosphosites, of which 669 phosphosites were significantly different between HFD and LFD mice livers. We detected alterations of proteins associated with cellular metabolic processes such as small molecule catabolic process, monocarboxylic acid, long- and medium-chain fatty acid, and ketone body metabolic processes, and peroxisome organization. We observed a significant downregulation of protein phosphorylation in HFD fed mice liver in general. Untargeted lipidomics identified upregulation of triacylglycerols, glycerolipids and ether glycerophosphocholines and downregulation of glycerophospholipids, such as lysoglycerophospholipids, as well as ceramides and acylcarnitines. Analysis of differentially regulated phosphosites revealed phosphorylation dependent deregulation of insulin signaling as well as lipogenic and lipolytic pathways during HFD induced obesity. Thus, this study reveals a molecular connection between decreased protein phosphorylation and lipolysis, as well as lipid-mediated signaling in diet-induced obesity.

## 1. Introduction

Obesity is a prevalent health concern worldwide, and it is accompanied by a plethora of comorbidities. Among them, non-alcoholic fatty liver disease (NAFLD) is now recognized as the most common form of liver disease, affecting one quarter of the global population, which has a similar rate of prevalence as obesity [1]. NAFLD is characterized by fat accumulation in the liver that is not caused by alcohol consumption, and is associated with different factors, including an increase in dietary fat released from adipocytes via lipolysis, de novo hepatic lipogenesis, circulating free fatty acid (FFA) and a decrease in fatty acid oxidation [2,3]. NAFLD poses a significant health burden because even the earliest and most common type of NAFLD, simple steatosis, is shown to be not as prognostically benign as it has been thought for a long time. As an illustration, a recent large-scale cohort study of 10,568 patients with NAFLD in Sweden found that patients with simple steatosis showed a significantly elevated risk of extrahepatic cancer, hepatocarcinoma, cardiovascular disease and cirrhosis [4]. Therefore, it is critical to understand the molecular signatures involved in the onset of steatosis as the failure of proper liver functions will promote pathogenesis of metabolic complications in turn [5].

HFD feeding is advantageous for the establishment of fatty livers in mice as it generates less pronounced inflammation and rarer fibrosis after an extended period of intervention than in the case of a methionine-choline deficient diet [6,7]. Moreover, the model establishes pathologic phenotypes resembling human disease as it is accompanied by obesity, insulin resistance and hyperlipidemia [6,7]. Therefore, the diet-induced obesity (DIO) is typically associated with complex and intertwined metabolic abnormalities that primarily entail the involvement of a multitude of proteins, including their differential expression, post-translational modifications, and protein–protein interactions [8,9,10]. Similar to proteins, lipids also have extensive biological roles as signaling molecules, energy reserves and structural components of membranes. Gaining information about regulation/dysregulation of each class of lipids in response to DIO and integrating these results with proteomics and phosphoproteomics is important for our comprehensive understanding of the etiology of fatty liver. In the last three decades, significant progress has been made in our understanding of the wide-ranging changes in proteins, RNA and metabolites caused by obesity and overnutrition [8,9,11,12,13]. However, most of the omics studies on obesity and insulin resistance thus far have focused on one area (either proteomics or lipidomics), and only a few have focused on multi-omics approaches. Studies integrating global proteomics, phosphoproteomics and lipidomics of liver under the context of DIO are limited. Such integrated analyses, however, can be a powerful strategy to depict the changes in cellular functions controlled by highly connected protein and lipid molecules. This knowledge could be particularly useful for understanding the biology of obesity-related pathologies, and for the development of new treatment strategies. 

In the present study, we aimed to explore the changes in liver proteins and lipids and their functional role and relationship in the development of DIO at systems level by performing an integrative multi-omics analysis and correlating identified modules with DIO.

## 2. Materials and Methods

### 2.1. Mouse Husbandry and Diets

All procedures involving animals were performed in accordance with the National Institute of Health Guide for the Care and Use of Laboratory Animals and were reviewed and approved by the Purdue Animal Care and Use Committee (protocol number 1111000154). C57BL/6 male mice were kept in a humidity and temperature-controlled facility in a 12:12 h dark/light cycle with ad libitum access to food and water. From weaning to 5 weeks of age, mice were fed a chow diet consisting of 62.1% of calories from carbohydrate (starch), 24.7% from protein, and 13.2% from fat (PicoLab 5053, Lab Diets, Richmond, IN, USA). At 5 weeks of age, mice were randomly assigned to one of the two diets for an additional 12 weeks: low-fat diet (LFD, 10% of calories from fat, D12450J) or high-fat diet (HFD, 60% of calories from fat, D12492) (Research Diets, Inc., New Brunswick, NJ, USA). Body weight was recorded weekly. Fasting blood glucose was measured using the OneTouch glucometer (LifeScan, Milpitas, CA, USA). After 12 weeks, mice were fasted for two hours and euthanized by CO_2_ followed by cervical dislocation and livers were collected and stored at −80 °C for further analysis. Fat pad weight was measured at the time of euthanasia. 

### 2.2. Liver Tissue Preparation for Proteomics

Livers from six lean mice and six DIO mice were homogenized in 100 mM of ammonium bicarbonate, supplemented with protease and phosphatase inhibitors, in a Precellys Evolution, using CK14 soft tissue homogenizer tubes (Bertin Technologies SAS, Montigny-le-Bretonneux, France) for 3 × 20 s bursts at 6200 rpm. The protein concentration of each homogenate was determined by bicinchoninic acid assay kit (Thermo Fisher Scientific). An aliquot of each sample containing 500 μg of total protein was precipitated using ice-cold acetone at −20 °C overnight. After acetone removal, protein pellets were reduced with 10 mM dithiothreitol in 8 M urea, and alkylated using iodoethanol in ACN (2% iodoethanol, 0.5% triethylphosphine, 97.5% acetonitile). The proteins were digested with mass spec grade Trypsin/LysC mix (Promega, Wisconsin, WI, USA) at a 1:50 (*w*/*w*) enzyme-to-substrate ratio, using a barocycler (Pressure BioScience Inc., Easton, MA, USA) at 50 °C with 60 cycles of 20 kpsi for 50 s and 1 atmospheric pressure (1 ATM) for 10 s, following the Hedricks et al. protocol [14]. Samples were cleaned using Pierce Peptide Desalting Spin Columns (Thermo Fisher Scientific, Waltham, MA, USA). Samples were then divided into two tubes containing 50 μg and 450 μg peptides. The 450 µg peptides were further processed for phosphopeptide enrichment by using the PolyMac phosphopeptide enrichment spin-tips (Tymora Analytical, West Lafayette, IN, USA), following the manufacturer’s recommendations. Finally, samples were dried and resolubilized in 20 μL 0.1% formic acid (FA) in 3% acetonitrile (ACN) and 1 μg was loaded into the column for global profiling. Purified phosphopeptides were re-suspended in 10 μL of 0.1% FA in 3% ACN and 5 μL for phosphoproteomics.

### 2.3. Mass Spectrometry Analysis of Liver Proteome

Samples were analyzed by reverse-phase LC-ESI-MS/MS system using the Dionex UltiMate 3000 RSLC nano System, coupled to a Q-Exactive High-Field (HF) Hybrid Quadrupole Orbitrap MS (Thermo Fisher Scientific) as described previously [13]. Briefly, peptide separation was accomplished using a trap column (300 μm ID × 5 mm) packed with 5 μm 100 Å PepMap C18 medium, and then using a reverse phase analytical column (50-cm long × 75 µm ID), packed with 2 µm 100 Å PepMap C18 silica (Thermo Fisher Scientific, Waltham, MA, USA). The column was maintained at 50 °C, mobile phase solvent A was 0.1% FA in water, solvent B was 0.1% FA in 80% ACN and the loading buffer was 0.1% FA in 2% ACN. Peptides were solubilized in 0.1% FA in 3% ACN and loaded into the trap column in the loading buffer for 5 min at 5 μL/min, then separated with a flow rate of 150 nL/min using a 224.9 min linear gradient from 2% to 30% B, then changing to 60% B at 260 min before reverting to 2% B for re-equilibration for 20 min. The mass spectrometer was operated in positive ion and standard data dependent acquisition mode. The spray voltage was set at 2.6 kV, the capillary temperature was 320 °C and the S-lens RF level was set at 50. The resolution of Orbitrap mass analyzer was set to 120,000 and 15,000 at 200 *m/z* for MS1 and MS2, respectively, with a maximum injection time of 100 ms for MS1 and 20 ms for MS2. The full scan MS1 spectra were collected in the mass range of 350–1600 *m/z* and the MS2 first fixed mass was 100 *m/z*. The automatic gain control (ACG) target was set to 3 × 10^6^ for MS1 and 1 × 10^5^ for MS2. The fragmentation of precursor ions was accomplished by higher energy C-trap collision dissociation (HCD) at a normalized collision energy setting of 27% and an isolation window of 1.2 *m/z*. The data dependent acquisition settings were for top 20 MS2 with a minimum intensity threshold of 5 × 10^4^ and a minimum AGC target of 1 × 10^3^. The dynamic exclusion was set at 15 s and accepted charge states were selected from 2 to 7 with 2 as a default charge. The exclude isotopes function was activated. 

### 2.4. Data Analysis

LC–MS/MS data were processed with MaxQuant software (Ver 1.6.17.0) [15,16]. Raw spectra were searched against the mouse UniProt FA Mus musculus protein database (June 2021). Six amino acids were set as the minimum length required in the database search. The search was performed with a precursor mass tolerance of 10 ppm and MS/MS fragment ions tolerance of 20 ppm. Trypsin and LysC were set as specific enzymes, with up to two missed cleavages allowed. Oxidized methionine, and for the phosphoproteomics, phospho STY were defined as a variable modification, and iodoethanol of cysteine was defined as a fixed modification. The “unique plus razor peptides” were used for peptide quantitation and the false discovery rate of peptides spectral match and protein identification was set at 1%. “Label free quantitation” (LFQ) and “Match between runs” were enabled. Subsequent bioinformatic analysis was performed with Perseus (version 1.6.7.0) [17]. Proteins labeled “only identified by site”, “reverse”, or “contaminants” were removed from the analysis. Proteins were filtered for at least five valid values among six biological replicates (70%) in either of the conditions. Missing values were replaced by values derived from a normal distribution. Significantly up- or downregulated proteins between the two groups were determined by a Student’s *t*-test with Permutation-based FDR (q-value < 0.05, |log_2_(fold-change)| > 0.38). For phosphoproteomics, only class one phosphosites (localization probability > 0.75) were considered for downstream analysis. Functional annotations and enrichment analysis were performed using Metascape. Volcano plots were generated with the EnhancedVolcano R package [18]. 

### 2.5. Lipid Extraction and Measurement

Lipids were extracted using the Bligh and Dyer extraction method [19]. Briefly, 200 μL of liver homogenates with an equal amount of protein were transferred to a new tube and mixed with 550 μL of methanol and 250 μL of chloroform. The solution was vortexed briefly and incubated at 4 °C for 15 min, 250 μL of ultrapure water and 250 μL of chloroform were added and the sample was centrifuged for 10 min at 16,000× *g*, forming a 2-phase solution. Bottom organic phase was transferred to a new tube and dried and samples were stored at −80 °C until analysis. Lipid extracts were analyzed in an Agilent 1290 Infinity II UPLC, coupled to an Agilent 6545 quadrupole time-of-flight tandem mass spectrometer. Samples were resuspended in 30 μL of a mixture of ACN: methanol: water (3:5:2 *v*/*v*) and 8 μL were loaded to a Waters ACQUITY UPLC^®^ BEH C18 1.7 µm column with a controlled temperature of 45 °C. The binary pump used 10 mM ammonium acetate in water with 0.1% FA for mobile phase A and 10 mM ammonium acetate in a 50% isopropyl alcohol: 49.9% ACN: 0.1% FA for mobile phase B, at a flow rate of 0.4 mL/min. The liquid chromatography gradient was of 35% B at 0 min, 80% B at 5 min and ramped up to 100% B at 10 min, with a 5 min hold, then returned to 35% B in 2 min and a 4 min hold. The mass analyzer was set with a ESI capillary voltage of 35,000 Vcap, a sheath gas temperature of 320 °C and flow of 8 L/min, a nebulizer gas pressure of 35 psig, a fragmentor of 135 Vs and skimmer of 35 V. Mass spectrums were collected in profile mode with a range from 100 to 1200 *m/z* at a scan rate of 5 spectra/s with 200 min/spectrum for MS1 and 3 spectra/s with 333.3 mS/spectrum for MS2. Raw data were processed using MS-dial 4.7 [20] with the MSP spectral kit of 13,303 unique compounds in positive mode. 

A second lipid extract obtained as described above was used for fatty acid methylations (FAMEs) to analyze free fatty acid content by GC–MS. Before derivatization, the samples were spiked with 1 µg of C17:0 as internal standard. The samples were derivatized with 500 μL of 14% boron trifluoride solution (Sigma-Aldrich # B1252) and reacted for 30 min at 60 °C, followed by the addition of 500 μL of water and 500 μL of hexane. After mixing, 0.2 g of anhydrous sodium sulfate was added to the sample, and it was allowed to sit. The hexane layer was collected and dried. GC–MS derivatized samples were resuspended in 100 μL of 100% hexane in a Thermo Fisher Triplus RSH auto sampler and Trace 1310 gas chromatography (GC) system, coupled to a Thermo Fisher TSQ 8000 mass spectrometer (MS) (Thermo Fisher Scientific, Waltham, MA, USA), with an Agilent Select FAME GC column (50 m × 0.25 mm, film thickness 0.2 um) (Agilent Technologies, Santa Clara, CA, USA). The GC carrier gas was helium with a linear flow rate of 1.5 mL/min. The GC temperature gradient started at 70 °C at 0 min and was ramped up to 310 °C at 7 °C/min and held for 1 min for a total run time of 38.28 min. The GC inlet was set to 250 °C and samples were injected in split mode with a ratio of 10 and a flow of 15 mL/min. The MS transfer line was set to 250 °C and the MS ion source was set to 250 °C. MS data were collected in selected ion monitoring (SIM) mode. Raw data were analyzed with Thermo Fisher Chromeleon (Version 7.2.9) software and a standard mixture of 37 FAME (Sigma-Aldrich Corp., St. Louis, MO, USA) was used to confirm spectra and column retention times.

## 3. Results

### 3.1. Mice Fed a Chronic High-Fat Diet Develop Obesity

Male C57BL/6J mice were fed either an LFD or an HFD for 12 weeks starting at 5 weeks of age. After only 1 week, the body weight of the HFD fed group started to be significantly greater than that of the LFD fed group (*p* < 0.05) and the difference became greater throughout the 12 weeks. After 12 weeks of the diets, we confirmed that the HFD induced mice obesity (weight gain of 238%, Figure 1A). The HFD also increased adipose tissue fat deposition and elevated fasting blood glucose concentrations, which indicate defects in blood glucose control (Figure 1B,C).

### 3.2. Liver Proteome Profile Change upon Diet-Induced Obesity

To characterize the holistic changes in the proteome of murine livers during DIO, we performed a global proteomic analysis (Figure 2). In total, 29,590 peptides were identified, which could be assigned to 2973 proteins. We then filtered for proteins with LFQ values greater than zero in at least five out of six replicates in one group. After applying our filtering criteria, 2447 (82%) quantified proteins were retained for further analyses (Appendix A Appendix A), suggesting high reproducibility and reliability of the proteomic analysis, which was also showcased by distinct clustering in a PCA analysis (Figure 3A). The clear distinction between the changes in protein levels is further evidenced by a heatmap visualization of the significantly regulated proteins in our dataset, in which two hierarchical clusters are evident (Figure 3D). We then performed a Student’s *t*-test to unveil proteins that were significantly regulated after DIO. A volcano plot representation of our filtered data shows that the distribution of upregulated and downregulated proteins was remarkably symmetrical, with 312 proteins above the q < 0.05 and 0.38 |log_2_(fold-change)| cutoffs [21,22,23,24,25], set as the minimum to consider proteins as “significantly regulated” (Figure 3B). Specifically, 141 proteins were downregulated and 171 were upregulated. To gain an insight on the proteins that were most affected by DIO, we separately plotted all significant proteins with an absolute log_2_(fold-change) value greater than 2 (Figure 3C). We observed 12 downregulated proteins and 10 upregulated proteins, many of which are involved in lipid metabolism. Particularly, we observed that Vanin-1 (Vnn1), a glycosylphosphatidyl inositol-anchored protein was among these highly expressed proteins, which has been described as peroxisome proliferator-activated receptor alpha (Pparα) target gene, and upregulated in mouse fatty livers [26,27,28]. Recent study elucidated the role of Vnn1 in overactivation of gluconeogenesis, contributing to hyperglycemia [28], and our study also supports that Vnn1 may be a therapeutic target for NAFLD and associated glucose dysregulation. Lgals1 (galectin-1) was also highly upregulated in the livers of DIO mice (Figure 3C), which is in agreement with a previous study that showed increased Lgals1 expression in obesity [29]. Indeed, pharmacological inhibition of Lgals1 was shown to alleviate adiposity in obesity [30,31]. However, its role in fatty liver disease has not been as extensively investigated as galectin-3 [32,33,34], which was not identified in our proteomic analysis, warranting further investigation.

Thus, to categorize and characterize the proteins significantly regulated during DIO, we performed a Gene Ontology (GO) enrichment analysis of both upregulated and downregulated proteins together. Notably, several GO terms were listed as both downregulated and upregulated, such as monocarboxylic acid and metabolic process, small molecule biosynthetic process, sulfur compound metabolic process and others, which likely indicate a wholistic metabolic reprograming of such processes where specific subsets of proteins are modulated in distinct ways (Figure 4A). For instance, five of the ten proteins with the greatest fold change increase in the livers of DIO mice are involved in monocarboxylic acid metabolic process (GO:0032787): Cyp2b9; Cyp4a14; Gsta1; Acot3; and Vnn1, while another protein in the same GO term, Fabp5, is one of the proteins with the greatest fold change downregulation (Figure 3C). To easily visualize unique and common pathways in network format, nodes were shown as a dotted circle in Figure 4B.

Based on the TRRUST database, constructed with literature-curated human transcription factor–target interactions [35], we uncovered transcription factors associated with protein landscape change (Figure 4C). The result indicates that Pparα and Nrf2 (Nfe2l2) may contribute to DIO-induced protein upregulation, while Hnf4α, ChREBP (Mlxipl), Srebp1 (Srebf1) may contribute to DIO-induced protein downregulation at their transcriptional level. The enrichment of the transcription factor Pparα in response to an HFD is in line with the observation that Vnn1 is one of the proteins with the highest increase in its protein levels, considering that Vnn1 is a key regulator of Pparα in the liver [27].

### 3.3. Differentially Expressed Phosphoproteins in the Liver from HFD and LFD Fed Mice

Protein phosphorylation is a major driver of protein function, protein localization and protein–protein interactions [36,37]. Thus, to gain an insight on how proteins are regulated during DIO in a larger scope, which transcends the changes in protein levels alone, we performed a comprehensive phosphoproteomic analysis of the liver tissues in which the global analysis was performed. We identified a total of 1391 phosphoproteins, containing 1650 class I phosphosites (phosphosites with localization probability > 0.75). We also observed 14 novel phosphosites on the proteins Eif3m, Ubxn, Atf7, Reps2, Gprin3, Tacc2, Igf1r, Gsr, Ubxn7, Snx11 and Abcb11 (Appendix A Appendix A). Of the 1650 class I phosphosites, 1517 (92%) sites were serine, 124 (7.5%) sites were threonine and 9 (0.5%) were tyrosine phosphorylated. Of these 1650 phosphosites, 669 sites were significantly different between HFD and LFD fed mice; 15 were upregulated and the remaining were downregulated (Appendix A Appendix A). This result suggests a significant decrease in site-specific phosphorylation in HFD fed mice, highlighting the changes in protein regulatory mechanisms in mice due to DIO. Increased phosphosites in HFD were observed in transporter proteins such as sodium bicarbonate transporter (Slc4a4), solute carrier anion transporter (Slco1b2), helicase ARIP4 (Rad54l2), RNA binding protein (Rbm39), and glycerol-3-phosphate acyltransferase (Agpat9). Of the nine tyrosine phosphorylation sites, six were significantly different with one increased (Slc4a4) and five decreased (Gsk3a, Gpxl, Uox, Mapk14 and Arhgap35) phosphorylation levels. Further, five of the nine tyrosine phosphorylation sites belonged to proteins containing known kinase domains (Mapk1, Mapk14, Gsk3a and 3b, Dyrk1a and 1b, and Prpf4b; Appendix A Appendix A), and the phosphorylation levels of all these sites decreased in HFD mice liver. We also identified an additional six phosphoproteins with known kinase domains, and all showed decreased phosphorylation at serine and threonine residues. A predominant decrease in phosphorylation is easily visualized as a heatmap (Figure 5A) and further evidenced by a volcano plot, representing all filtered phosphosites, in which samples corresponding to HFD are marked by the consistent downregulation of thousands of phosphosites (Figure 5B).

Several solute career (Slc) transporters including Slc2a2, Slc10a1, Slc16a1, Slc16a10, Slc16a7, Slc33a1, Slc38a4, Slc4a1, Slc4a4, and Slc4a10 had downregulated phosphosites in HFD fed mice liver (Appendix A Appendix A). Although several of these transporters were also downregulated at the protein level, their phosphosite level downregulation was greater. These transporters facilitate the transport of substrates across cell membranes, including glucose, inorganic and organic ions, small molecule drugs, xenobiotics and amino acids, and contribute to insulin signaling, glucose homeostasis and the etiology of different metabolic diseases [38,39]. The glucose transporter isoform, Glut2 (known as Slc2a2), was phosphorylated at serine 522, and phosphorylation decreased in HFD compared to LFD mice liver (Appendix A Appendix A). Glut2 expression was also down at the protein level, suggesting impaired glucose homeostasis and dysregulated insulin response. Other notable downregulated solute transporters were lactate and H^+^ transporter, Slc16a1; acetyl-CoA transporter, Slc38a1; sodium and amino acid transporter, Slc33a1; and sodium and bile acid transporter, Slc10a1. We also identified downregulation of acetyl-CoA carboxylase (Acaca and Acacb), both at the protein and phosphosite levels (Appendix A Appendix A), and again, downregulation at the phosphorylation level was greater than at the protein level. Acaca was phosphorylated at serine 23, 29 and 79, whereas Acacb was phosphorylated at serine 1332. The decreased phosphorylation of the enzymes in all these phosphosites under the HFD fed condition further indicates dysregulation of lipogenesis and glucose homeostasis.

GO enrichment analysis of proteins regulated at their phosphosite levels indicated that RNA metabolism was highly affected by HFD. Interestingly, we also found that cellular response to insulin stimulus was one of the top 20 most significant downregulated processes among the enriched phosphoproteins (Figure 5C). Many other biological processes, as expected from previous studies on diet-induced obesity [8,40,41], are related to cellular metabolic and catabolic processes, membrane and organelle organization, cell–cell communication, protein localization, apoptotic signaling, and oxidative stress responses (Figure 5C).

Recent studies have reported that the tumor suppressor, promyelocytic leukemia (PML) protein, plays a regulatory role in cellular metabolism, by controlling Ppar, which, in turn, is involved in an important signaling pathway that modulates lipid homeostasis [40,41]. PML is known as the key organizer of the PML-Nuclear Bodies (PML-NBs), and has fascinated scientists for many years due to its multifaceted role under many cellular conditions, notably acute promyelocytic leukemia [42,43]. Liver PML ablation induces extensive reprograming of metabolic pathways, including an accelerated fatty acid metabolic rate accompanied by decreased total lipid accumulation in the liver, as well as insulin resistance [44]. To investigate the effects of HFD in PML-NB proteins, we constructed a library using protein–protein interaction databases and filtered for the proteins identified in our global and phosphoproteomics study. Many well-known PML-NB proteins were identified and were regulated at their protein and/or phosphosite levels (Figure 5D), including the protein PML. Phosphorylation of PML was downregulated, notably at S17, a phosphosite targeted by the kinase Cdk. (Appendix A Appendix A). Further, phosphosites belonging to several E3 sumo and ubiquitin ligases were also identified and were downregulated in the HFD mice liver compared to the LFD mice liver, including RanBP2, Nedd4, Praja-1, Rbbp6, Urb4, and Zfp19 (Figure 5D). These results suggest that not only protein phosphorylation, but protein SUMOylation and ubiquitination may also be regulated in response to DIO. It is important to mention that decreases in the phosphorylation of the majority of the PML-NB proteins was independent of the relative changes in protein abundances.

### 3.4. Differentially Expressed Lipids in the Liver from HFD and LFD Fed Mice

Proteome and phosphoproteome analysis alone might not reflect the changes in the lipid content of the liver. To directly investigate the changes in lipid composition and regulation at the metabolite level due to HFD, we conducted an untargeted global lipidomics using LC–MS and GC–MS analytical systems. Untargeted lipid profiling detected a total of 3728 lipid ions present in at least 80% of one group. Data processing using MS-Dial allowed for the tentative identification of 1801 features after blank filter. A total of 464 lipid ions had MS2 acquired data and were used for statistical analysis (Appendix A Appendix A). Most lipids detected were triacylglycerols (TAGs) and phosphatidylcholines (PCs), followed by diacylglycerols (DAGs) and sphingolipids. Several lipids were suggested with Riken IDs and grouped as unknowns. FAMEs (hereon called FFA) were identified using GC–MS and normalized to C17 internal standard (Appendix A Appendix A). Lipid data were centered at the mean and divided by the standard deviation of each variable to scale them and account for the difference in intensities in the subsequent analysis.

Unsupervised PCA analysis of the lipid features’ relative intensities revealed clear separation of the two groups with 42.4% of explained variance (Figure 6A). The heatmap illustrates the difference in the detected lipid profile of each sample and shows two main clusters with opposite trends (Figure 6B). Univariate analysis comparing the lipid profiles of HFD and LFD fed mice identified 109 lipid features with *p*-values below 0.05 after the *t*-test, and 42 features with reduced and 67 features with increased relative amounts of at least 2-fold-change, which are represented in the volcano plot (Figure 6C). As expected, TAGs and FFA were increased in the HFD compared to the LFD and most PCs, PE and LPCs were decreased. Lipid ontology (LION) enrichment [45] was performed for lipid class overrepresentation analysis (Figure 6D), and showed upregulated TAGs, followed by glycerolipids and ether glycerophosphocholines. On the other hand, glycerophospholipids, including lysoglycerophospholipids and ceramides were significantly downregulated. Although not overrepresented, unusual acylcarnitines CAR (24:6) and CAR (28:6) were significantly downregulated. Glycerophosphoglycerols, such as BMP (35:4) and HBMP (48:0), were significantly reduced in the HFD compared to the LFD despite not being largely represented. LION analysis for cellular component, function, and physical and chemical properties (Figure 6E), showed that lipid storage and lipid droplet formation associated lipids were significantly increased in DIO mice, as well as lipids with neutrally charged head groups. In addition, 12 to 22 carbon fatty acids either saturated, monounsaturated or with more than three double bonds were increased, while 24 to 26 carbon fatty acids with two and three double bonds were reduced compared to LFD. Lipids linked to endoplasmic reticulum (ER) and lipid-mediated signaling were significantly reduced in the HFD compared to the LFD, as well as lipids associated with the membrane components, intrinsic curvature and positive/zwitterion head group.

To derive insights on whether distinct lipid components align with the changes in proteins or phosphoprotein profiles, correlations between detected lipid species and significantly changing proteins in the global and phosphoproteomics were analyzed. Lipids were assigned to their respective classes, and the class correlation was obtained as described by Chauhan et al. [46]. Variable importance in projection (VIP), which is defined as the weighted sum of squares of the loadings in a partial least squares discriminant analysis, that takes into account the amount of explained variation in each dimension, was used to rank the features. Correlation heatmaps of global proteins and lipid classes showed that glycerophosphoglycerols, ether-linked digalactosyldiacylglycerols, LPC, LPE, neutral sphingolipids, and PS presented stronger correlation (*p* < 0.0 and VIP > 0.8) with the identified proteins, while triacylglycerols did not appear highly correlated with global proteins (Figure 7A; Appendix A Appendix A). When examining the phosphoproteomics correlation heatmap (Figure 7B; Appendix A Appendix A), fewer lipid classes had significant correlation across the phosphosites identified. Ether-linked PC (EPC) and FFA were predominantly negatively correlated with phosphosites, such as Tnks1bp1 (S796), Chd4 (S508), Grb7 (S420), Acss2 (S263), and Bad (S155), while acylcarnitines, glycerophosphoglycerols, LPC and LPE were mostly positively correlated, with the exception of Thrap3 (S669), Agpat9 (S68), Hadh (S13), Pdha1 (S300), Prpf4b (S145), Rad54l2 (S1168 and S1171), Rbm39 (S127 and S129), Rpap3 (S429), Scaf4 (S154), Slc4a4 (S68), Slco1b2 (S290), Sord (S169), Srsf2 (S189 and S191), and Tnks1bp1 (S866) (Figure 7B). To gain insights on the potential biological implications of strong correlation between certain lipid classes and phosphoproteins, significantly different proteins with a VIP larger than 1 that are in significant correlation with lipids were analyzed using REACTOME for pathway analysis including interactors. The identified significant pathways were comprised in the main categories of metabolism, programmed cell death, signal transduction, transport of small molecules, disease, and metabolism of proteins. Different pathways involved in energy metabolism were highlighted such as glycolysis, gluconeogenesis, ChREBP activates metabolic gene expression, and PKA-mediated phosphorylation of key metabolic factors, as well as PP2A-mediated dephosphorylation of key metabolic factors (Figure 8A). Pathways related to mitochondria function including activation of PPARGC1A (PGC-1α) by phosphorylation and carnitine metabolism were also significantly enriched (Figure 8A). Several proteins were grouped under translation by ribosomal scanning and start codon recognition and signaling by Rho GTPases (Figure 8A). Interestingly, when individual phosphorylation sites classified under certain pathways were categorized by correlation coefficient with each lipid class (Figure 8B), it was evident that not all the phosphosites had the same behavior. For example, Eif3b had four different phosphosites that meet the correlation cut-off criteria, but each one of those sites showed different correlations with AC, FFA, Ether-PC, LPC and LPE, highlighting the specificity of phosphosites and their unique regulation within a protein. Another similar example is membrane protein Slc4a4. Slc4a4 (S68) had a negative correlation with phospholipids, but such a relationship was not observed with Slc4a4 (Y64).

## 4. Discussion

Applying the multi-omics approach, we identified significant modulation of the liver proteome and phosphoproteome involved in signaling, implicated in obesity and insulin resistance. We observed decreased phosphorylation of the majority of identified phospho STY sites in HFD fed mice, suggesting that metabolic disorders due to HFD induced obesity are characterized by dysregulated phosphorylation-dependent signaling. Furthermore, decreased site-specific phosphorylation of many protein kinases and key metabolic enzymes involved in lipid and glucose homeostasis imply underlying functional consequences of obesity and insulin resistance.

### 4.1. Altered Proteomes Related to Lipoprotein Assembly, Lipoprotein Uptake, De Novo Lipogenesis (DNL) and Fatty Acid Uptake

The liver predominantly produces lipoproteins, other than chylomicron. DIO animals had abundant apolipoproteins (ApoA4, ApoC1, ApoC2, ApoC4, ApoE, ApoH), which indicates an increased level of circulating lipids. In terms of cellular uptake of blood lipids, fatty acid uptake protein Cd36 was highly abundant in the livers of obese animals. A previous study showed Cd36 null mice are protected from hepatic steatosis induced by Lxr agonists and that Cd36 is a common target of Lxr, as well as Pparγ and pregnane X receptor [47]. Although we did not identify Pparγ in global proteomic analysis, TRRUST analysis revealed Pparγ as an enriched nuclear receptor that could explain significantly upregulated proteins in DIO animals (Figure 4C). Furthermore, Lgals1, whose overexpression increases Pparγ protein level and transcriptional activation [29], was one of the most upregulated proteins in DIO animals (Figure 3C) and was positively correlated with FFAs having the highest VIP value. Evidence collectively suggests that the activation of Pparγ played a major role in altered protein abundance involved in lipid metabolism in DIO.

Three key rate-limiting enzymes in de novo lipogenesis (DNL) (Acly, Acaca, Fasn), as well as Scd1, that converts saturated fatty acids into monounsaturated fatty acids in the final step of DNL, were present in lower amounts in the livers of the obese mice. Acly was also found to be decreased in terms of phosphorylation at Ser455 in obese animals, and the change in phosphorylation at this site was greater than the changes at its protein levels, adding another layer of DNL attenuation, as phosphorylation at Ser455 is known to increase Acly enzymatic activity [48]. On the other hand, despite a decreased Acaca protein level (log_2_(fold-change): −0.83677), diminished phosphorylation at the regulatory site Ser79 was observed at a greater extent (log_2_(fold-change): −1.289). This suggests that Acaca is likely in an enzymatically active state, despite the decrease in its protein levels in DIO, as phosphorylation is known to inhibit Acaca activity [49].

Other lipogenic enzymes such as desaturases that introduce double bonds in acyl chains (Fads1, Fads2) and elongases which elongate C16 to longer acyl chains (Elovl2, Elovl5) were identified, but did not show a difference between the lean and obese animals. Meanwhile, lipid droplet proteins (Plin2, Plin4) were present in higher amounts in obese animals, indirectly showing heightened lipid storage. Hepatic DNL is mainly controlled by transcriptional regulation of genes by Srebp1 [50], Chrebp [51], liver X receptor α [52], and Pparγ [53]. In agreement with these findings, Srebp1 and Chrebp were shown as enriched transcription factors responsible for downregulated proteins, according to our TRRUST analysis (Figure 4C). Overall, lipid accumulation in the livers of HFD fed animals is largely attributed to an increased lipid flux rather than DNL.

Impaired glucose balance and insulin resistance characterize metabolic disorders in DIO. There is growing evidence that Slc transporters contribute to the etiology of various metabolic diseases [54]. Slc transporters are located in membranes and are highly expressed in the liver, kidneys, heart, gut, and brain, and are emerging as potential drug targets [39]. They serve as a ‘metabolic gate’ in cells and mediate the transport of a wide variety of nutrients and metabolites such as glucose, amino acids, vitamins, neurotransmitters, inorganic ions, organic anions, metals, and amino acids. Among the 35 identified Slc transporters, the expression of 12 members showed significant changes between the HFD and LFD fed mice liver, of which 2 members (Slc22a1 and Slc25a10) had increased expression and 10 (Slc10a1, Slc25a11, Slc25a20, Slc25a22, Slc25a3, Slc2a2, Slc33a1, Slc6a13, Slc9a3r1, and Slc01a1) had decreased expression. We identified several of these Slc transporters with diminished site-specific phosphorylation in the HFD. This may suggest defects in insulin signaling and transport of glucose and other molecules to membranes, essential for normal cellular function. The transporter, Slc2a2, is responsible for the transport of glucose into β-cells and facilitates glucose-mediated insulin secretion and signaling. Recessive mutation of the Glut2 gene in mice showed hyperglycemia and abnormal glucose homeostasis [55]. Its downregulation and loss of phosphorylation in our study further suggest the impaired utilization of glucose. However, the functions of other identified Slc transporters and the consequences of their loss of phosphorylation in obesity are unknown.

### 4.2. Increased Mitochondrial Fatty Acid β-Oxidation, Ketone Body Formation

Intracellular fatty acids are activated by acyl-CoA synthetase before being channeled into different metabolic fates, such as β-oxidation, triacylglycerol synthesis or phospholipid synthesis. Our results show Acsl1, a predominant isoform in the liver, was present in the obese animals, increased below our cut-off threshold (log_2_(fold-change): 0.306). In sharp contrast, Acsl4 was significantly less abundant in the obese animals (log_2_(fold-change): −1.478). Previous studies have shown arachidonic acid is a preferred substrate for Acsl4 [56] and that Acsl4 protein expression is controlled by substrate induced post-translational regulation, in which arachidonic acid promotes ubiquitin-proteasomal degradation [57]. Our fatty acid measurement (Appendix A Appendix A) shows a high level of arachidonic acid in the livers of DIO animals, and this may have contributed in part to reduced abundance of the protein. Other acyl-CoA synthetases with significant fold-change include Acss3 (log_2_(fold-change): 1.64) and Acss2 (log_2_(fold-change): −0.97). Acss3 is a mitochondrial enzyme whose expression is upregulated under ketogenic conditions [58], and our data show abundance in the livers of DIO animals. On the other hand, Acss2 was not only less abundant, but also showed lower phosphorylation levels (Ser30, Ser263, Ser267) in the livers of obese mice. Suppression of Acss2 in the liver after HFD feeding was reported previously [59], and others have reported mice lacking Acss2 are protected from steatosis induced by DIO [60]. This suggests that downregulation of Acss2 may serve as a defense mechanism against excess fat storage in the liver. However, it is unclear how phosphorylation affects enzymatic activity and future study is warranted.

Our results are consistent with previous reports on the proteomic landscape of hepatic steatosis by showing an abundance in proteins involved in mitochondrial β-oxidation. Carnitine shuttling enzymes (Cpt1α, Slc25a20) were more abundant in the livers of obese animals, transporting excess fatty acids from the cytoplasm into the mitochondrial matrix. Acadm and Acad11 enzymes that catalyze dehydrogenation of fatty acyl-CoAs to form enoyl-CoA were also more abundant in the obese animals, while the levels of Acadl were not significantly different between the two groups. Acads and Acadvl were present at a higher level in the livers of obese animals but did not meet our fold change cut-off value. The enzymes that catalyze the hydration of the enoyl-CoAs (Ech1, Hadh) are abundant in obese animals, with upregulation of Hadh Ser13 phosphorylation.

Hepatic ketogenesis occurs in the mitochondria and is activated to convert excess acetyl-CoA generated from β-oxidation into ketone body intermediates. Several proteins involved in ketogenesis were upregulated in the liver of the HFD fed animals: Bdh1; Acat1; Hmgcs1,2; and Hmgcl. Aacs, that converts ketone bodies into acetoacetyl CoA for cholesterol or fatty acid synthesis were less abundant in the HFD fed animals.

### 4.3. Increased Abundance of Proteins Involved in Peroxisomal β-Oxidation and Microsomal ω-Oxidation

Peroxisomes are single membrane-enclosed subcellular organelles, particularly abundant in hepatocytes. One dynamic metabolic process that peroxisomes participate in is the degradation of fatty acids that cannot occur in the mitochondria, specifically (1) very-long chain fatty acids with at least 22 carbons; (2) branched-chain fatty acids; (3) bile acid intermediates such as di- and trihydroxycholestanoic acids; (4) long-chain dicarboxylic acids [61]. Peroxisomal acyl-CoA oxidases that participate in catalyzing peroxisomal β-oxidation (Acox1, Acox2, Acaa1a, Acaa1b), as well as the bifunctional protein, Ehhadh, that catalyzes the hydration and dehydrogenation step of β-oxidation, were all upregulated in the obese animals. Acyl-CoAs, that are produced during peroxisomal β-oxidation, can be hydrolyzed into fatty acids by Acot4, or, alternatively, converted to carnitine esters and free CoA by Crat and Crot, all of which were more abundant in HFD fed mice. Accordingly, our results show that proteins involved in peroxisome biogenesis were abundant in the obese mice, suggesting increased demand for the maintenance and biogenesis of the organelle to support increased peroxisomal function. Peroxisome organization (GO:0007031) was enriched uniquely in the livers of DIO animals (Figure 4A) and especially, proteins involved in peroxisome biogenesis known as peroxins (Pex1, Pex11a, Pex16, Pex19, Pex3, Pex6) were more present in the obese animals.

Another lipid catabolism pathway that increased in response to elevated hepatic lipid overflow was microsomal ω-oxidation. The fatty acid ω-hydroxylases, Cyp4a10 and Cyp4a14, were highly upregulated in obese animals (Figure 3C). The resultant fatty ω-hydroxy acids are catalyzed by two subsequent reactions by alcohol dehydrogenase and aldehyde dehydrogenase. Our results show that microsomal fatty aldehyde dehydrogenase, Aldh3a2, was also more abundant in the obese animals. Overall, HFD feeding stimulated mitochondrial, peroxisomal and microsomal fatty acid oxidation systems. The fact that the key transcriptional factor, Pparα, is most enriched in DIO animals, according to our TRRUST analysis (Figure 4C), further supports upregulation of β-oxidation in three different subcellular organelles.

### 4.4. Decreased Glycolysis and Increased Gluconeogenesis

5′-AMP-activated protein kinase (AMPK) is an energy sensor kinase that activates energy-producing pathways when the intracellular ATP level is low. In the livers of obese mice, AMPK catalytic α-1 subunit, Prkaa1 (S496), had reduced phosphorylation, as well as non-catalytic gamma subunits, Prkag2 (S71, S87, S90) and Prkab1 (S108, S96), indicating reduced kinase activity. Prkag1 was also less abundant in the livers of DIO. Accordingly, we identified that several proteins under AMPK regulation showed reduced phosphorylation levels at specific sites in the obese animals. For example, glycogen synthase 2 (Gys2) phosphorylation at Ser8 (log_2_(fold-change): −1.7954) and Ser 627 (log_2_(fold-change): −4.591) was downregulated in DIO, indicating the active state of the enzyme, favoring glycogen synthesis despite a decreased protein level (log_2_(fold-change): −0.7757).

Meanwhile, decreased phosphorylation states of glycolytic enzymes that are under the control of AMPK induce glycolysis inhibition. Our data show downregulated phosphorylation of Pfkfb1 at Ser33, which indicates the activation of bisphosphatase and degradation of fructose-2,6-biphosphate, favoring gluconeogenesis [62]. Moreover, reduced phosphorylation of Pfkl at Ser775 further supports increased gluconeogenesis by decreasing kinase activity [63]. Carbohydrate response element-binding protein (ChREBP), which regulates gene expressions involved in glycolysis, gluconeogenesis and DNL, is one of the main transcription factors enriched in our downregulated protein dataset (Figure 3C). Consistently, our phosphoproteome results shows ChREBP phosphorylation was decreased at Ser514 in DIO, the residue critical for maintaining the transcriptional activity of ChREBP by enhancing ChREBP O-GlcNAcylation via phosphorylation [64]. Accordingly, one of the ChREBP target genes, Pklr, was less abundant in DIO animals, supporting decreased glycolysis.

### 4.5. Altered Abundance of Proteins Involved in ROS Scavenging

The influx of free fatty acids into hepatocytes leads to increased ROS, such as hydrogen peroxide (H_2_O_2_) and superoxide anion (O_2_^●−^) [65,66]. Our proteome results are consistent with previous studies which show the induction of antioxidant systems in the steatotic liver. Glutathione peroxidase 4 (Gpx4) is an enzyme found mostly in the mitochondria, which directly detoxifies membrane lipid peroxides at the expense of glutathione. Gpx4 was more abundant in the livers of HFD fed mice. Glutathione synthetase (Gss), that replenishes glutathione from glycine and γ-glutamylcysteine, was also upregulated in DIO. Furthermore, in our TRRUST analysis, Nrf2 was significantly enriched (Figure 4C). In DIO, the glutathione-S transferases (Gst), targets of Nrf2, including Gstm1,2,3,4, Gstk1, Gsta1,4 and Gstt3, were more abundant. Another Nrf2 target, Nqo1, was also more abundant in DIO animals. However, few Gst isoforms were less abundant in DIO: Gstm7; Gstp1,2; and Gstt2, making cellular modified amino acid metabolic pathway (GO:0006575) a common GO term in both the lean and DIO groups. A future investigation needs to identify the biological significance of differential induction of multiple Gst isoforms during DIO.

### 4.6. Regulation of PML-NB Proteins under HFD

Although the role of PML as a tumor suppressor protein is well known, its role in cellular metabolism, notably under obesity or HFD conditions, is unknown or inconclusive. Carracedo and Pandolf [67] reported increased hepatic PML protein levels during liver steatosis. The PML protein interferes with liver metabolism and controls fatty acid oxidation in stem cells [68], and the depletion of PML in mice decreases liver fatty acid accumulation after a long-term Western diet [44]. In contrast, other studies showed that PML inhibits adipogenesis, and loss of PML results in fat accumulation in mice [40]. It is possible that differences in a mouse strain, diets, aging, and environment may contribute to the inconsistent phenotypes [44,69]. Our results also emphasize the role of PML in diet-induced obesity and cellular metabolism, and its regulation by phosphorylation. This PML-dependent regulation may explain some of the inconsistencies reported in previous studies, as none of those studies investigated the hepatic function of phosphorylated PML. Downregulation of many E3 sumo and ubiquitin ligases also suggest the importance of these modifications, and further highlight the regulation of PML and other NB proteins via diverse modifications, as phosphorylation, SUMOylation and ubiquitination are known to be interdependent to each other to execute cellular functions under various stressed conditions [70,71].

## 5. Conclusions

Using the multi-omics approach, we showed how changes in lipid composition and quantity are correlated with the changes in the liver proteome and phosphoproteome. To the best of our knowledge, this is the first study to compare the relationships in terms of proteins, phosphoproteins, and lipids in mouse liver under HFD-induced obesity. Free fatty acids and ether glycerophosphocholines were negatively correlated with most of the significantly dysregulated phosphosites, while other lipid classes were positively correlated in general. This suggests the importance of the lipid environment for the membrane transporters and kinases functionality and highlights the relevance of future mechanistic studies in this domain. Although several interesting relationships were discovered between proteins and lipids, additional studies are necessary to determine the directionality of these relationships. Although the changes at the site-specific phosphorylation level were greater than the changes at the protein levels in the majority of cases, further studies will determine the degree to which such changes are due to direct site-specific phosphorylation changes that are independent of the changes at the relative protein abundances. One of the limitations of this study is the fact that no targeted analysis was performed for proteins, phosphoproteins, and lipids that provides confirmatory identification and absolute quantification data. Moreover, no mechanistic analyses were performed to determine the directionality of the findings. However, these results offer new opportunities for improving our understanding of the shift in liver metabolism of mice fed excess calories and provides data that could be useful for future investigations that aim to find new treatments to control the onset or slow down the progression of HFD induced hepatosteatosis.

## Figures and Tables

**Figure 1 cells-11-01172-f001:**
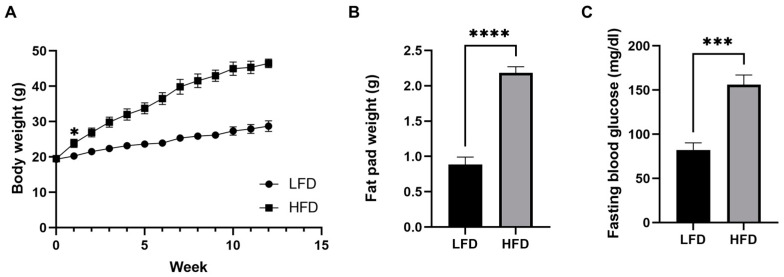
Diet-induced phenotypic variations in male mice. (**A**) Body weight over time. (**B**) Fat pad weight after 12 weeks of diet (*n* = 6/group, mean ± SEM). (**C**) Fasting blood glucose after 12 weeks of diet. Student’s *t*-test * *p* < 0.05, *** *p* < 0.001, **** *p* < 0.0001. Data shown as mean ± SEM. *n* = 6 LFD, *n* = 6 HFD.

**Figure 2 cells-11-01172-f002:**
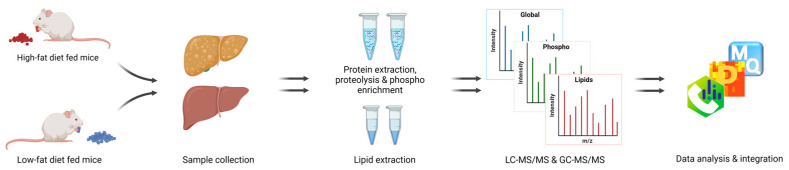
Workflow. Fatty liver establishment by HFD, followed by proteomic, phosphoproteomic, and lipidomic analysis.

**Figure 3 cells-11-01172-f003:**
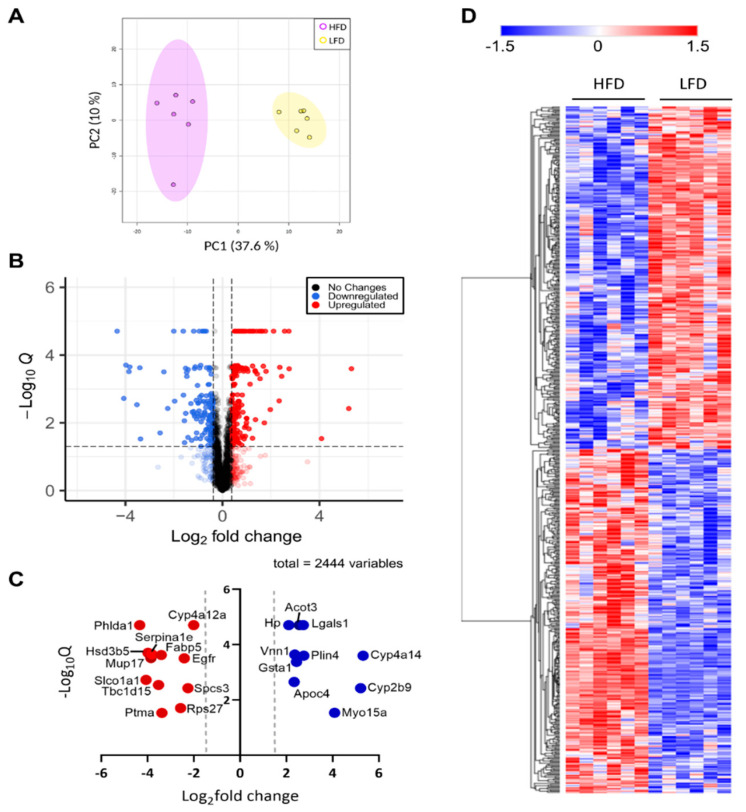
Global proteomic analysis. (**A**) PCA plot of significant proteins in each biological replicate. (**B**) Volcano plot of all quantified proteins. Zero q-values were converted to lowest non-zero q-value × 10^−1^ for this data representation. Blue color denotes downregulated proteins, red denotes upregulated proteins, and grey denotes proteins that do not change in response to administered diet. (**C**) Proteins with a *t*-test q-value < 0.05 and a |log_2_(fold-change)| > 2. Color legend is as described in (**B**). (**D**) Heatmap representation of all significant proteins clustered by hierarchical clustering. Color legend represents z-scores.

**Figure 4 cells-11-01172-f004:**
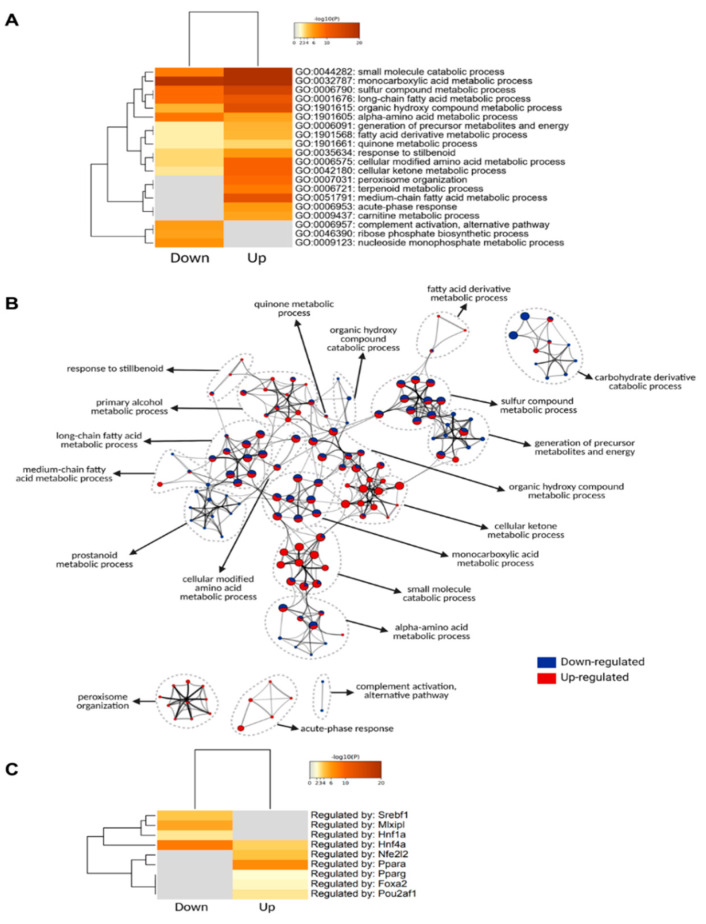
Gene ontology analysis of significantly regulated proteins. (**A**) Metascape enrichment analysis of statistically enriched GO Biological Process (BP) terms. The terms were selected with the best *p*-value within each cluster as its representative term. (**B**) Enriched clusters’ GO-BP terms in network format. Each enriched term is displayed as a circle node whose size is proportional to the number of input proteins that fall into the term. Each node is represented as a pie chart and each pie sector is proportional to the number of hits originated from either up- or downregulated proteins. (**C**) Enrichment of transcriptional regulators of significantly regulated proteins using TRRUST (transcriptional regulatory relationships unraveled by sentence-based text-mining) database.

**Figure 5 cells-11-01172-f005:**
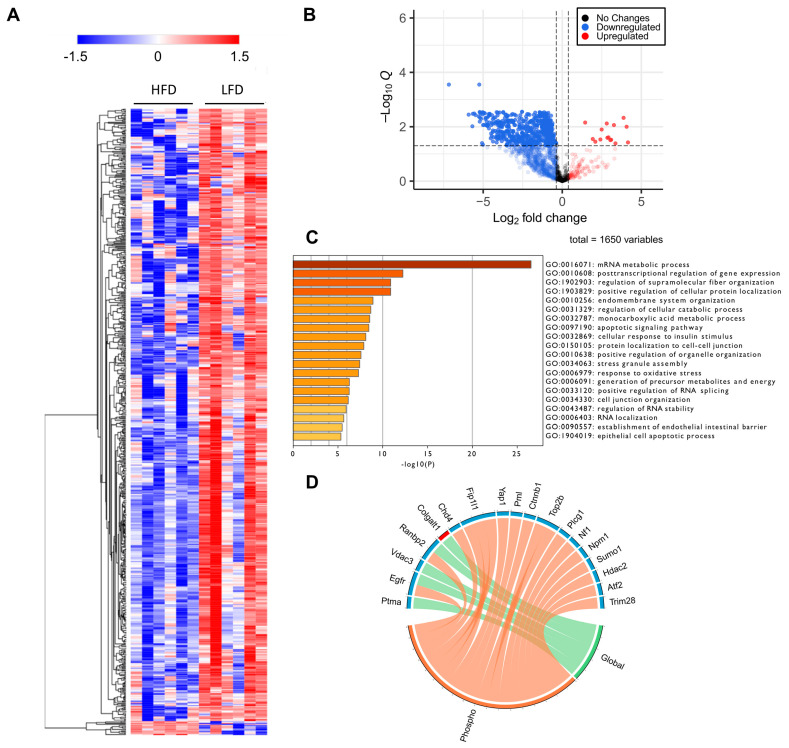
Phosphoproteomic analysis. (**A**) Heatmap representation of all significant phosphosites. Red indicated upregulated phosphosites and blue indicated downregulated phosphosites. (**B**) Volcano plot of all quantified phosphosites. Zero q-values were converted to lowest non-zero q-value × 10^−1^ for this data representation. (**C**) Gene Ontology for biological processes enriched from proteins with significantly downregulated sites. (**D**) PML-NB proteins significantly regulated at phospho and global levels. It is evident that many PML-NB proteins were significantly downregulated at the phosphorylation levels but not at the global level.

**Figure 6 cells-11-01172-f006:**
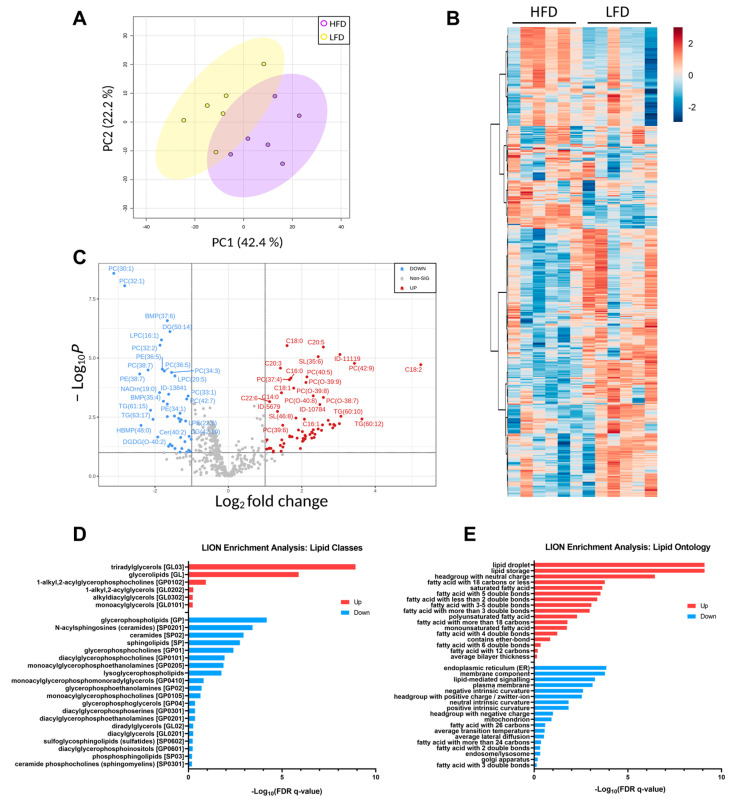
Lipid profiles. (**A**) PCA plot of each biological replicate. (**B**) Heatmap representation of all significantly changing lipids. Red indicated upregulated lipid species, and blue indicated downregulated lipid species. (**C**) Volcano plot of all quantified lipids. (**D**) Principal lipid classes changed in the two groups. (**E**) Lipid ontology enrichment for function, cellular component and chemical and physical properties significantly regulated.

**Figure 7 cells-11-01172-f007:**
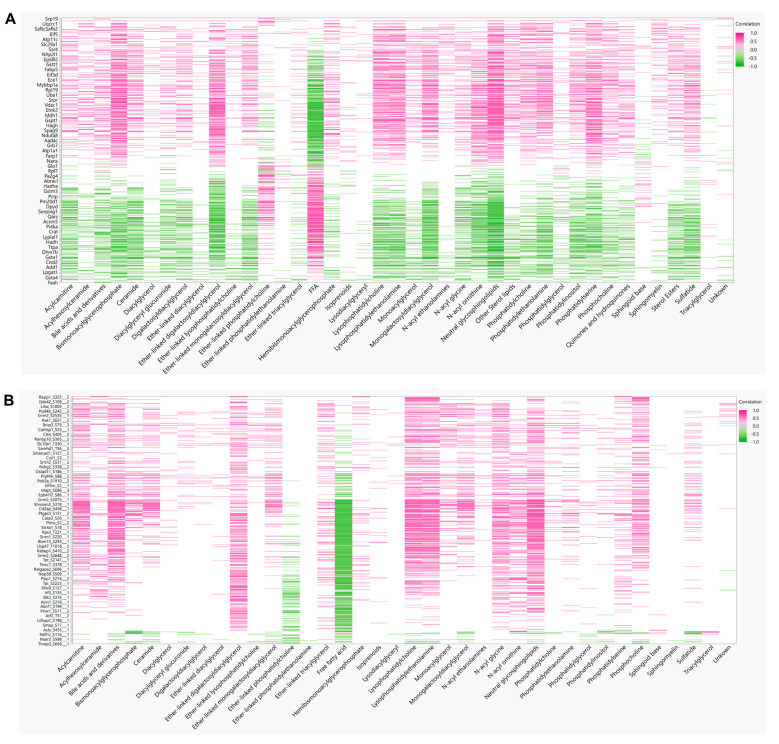
Protein–Lipid correlations. (**A**) Pearson correlation matrix between 38 lipid classes and 255 significant proteins, and (**B**) 649 significant phosphosites.

**Figure 8 cells-11-01172-f008:**
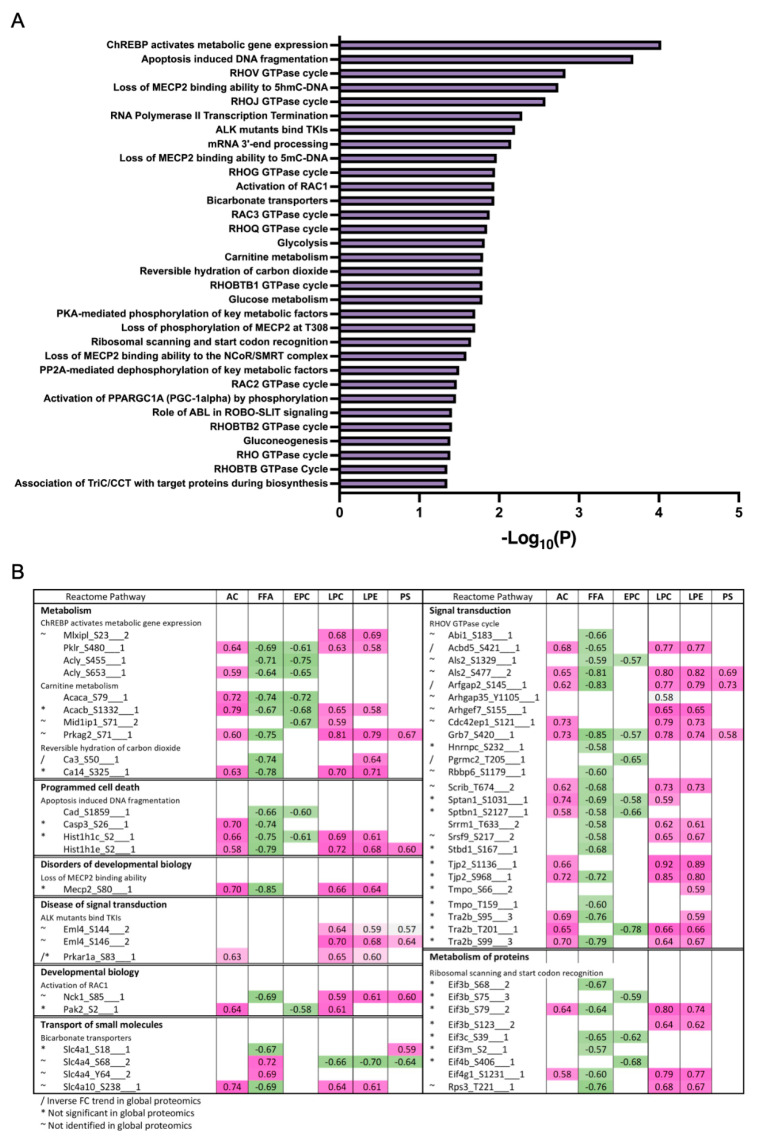
Reactome enrichment of correlation analysis for phospho datasets. (**A**) Reactome enriched pathways from global phosphoproteins significantly correlated with lipids. (**B**) Correlation analysis of phosphosites of interest, categorized by enriched reactome pathway and lipid classes.

## Data Availability

Raw LC-MS/MS data files can be accessed through MassIVE (https://massive.ucsd.edu/) with the ID: MSV000088835.

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
