# Peer review of "Multi-Omics Approach Reveals Dysregulation of Protein Phosphorylation Correlated with Lipid Metabolism in Mouse Non-Alcoholic Fatty Liver"

_cells, 2022, doi:10.3390/cells11071172_

Round 1
Reviewer 1 Report
Review of “Multi-omics approach reveals dysregulation of protein phos-2 phorylation correlated with lipid metabolism in mouse fatty 3 liver”
Firstly I find the subject matter of high interest, and suitable for submission to a high impact journal. I consider the idea of combinatorial proteomics/lipidomics also of high value. In general I found the paper to be well written, clear and legible in good English with only a few spelling errors. However I found some significant issues that must be addressed. My speciality is in proteomics, so the authors will find no assessment of the lipidomics work within this review, and would request the Editor to send the paper to another appropriate reviewer.
At this particular time, I am advising that this article undergoes significant revisions, outlined below.
- I find the title needs to be clearer as the subject at hand is non-alcoholic fatty liver disease, not just fatty liver.
- Line 75 of the introduction (“Our comprehensive analysis …”) to the end of the introduction should be removed. This should be in results or discussion. The introduction is no place for conclusions.
- Section 2.2 needs further explanation: from a proteomics perspective the sample preparation protocol is slightly unusual. Whereas in-solution digestion is commonplace, more modern concepts such as S-Traps and SP3 are now widespread. This in itself is not an issue, however the use of iodoethanol in place of the more common IAA or CAA needs explanation, perhaps? A reference to a publication where this method is described fully is needed. As is the use of the barocycler. Please cite these properly. Sample preparation in phosphoproteomics is an active field and I strongly advise the authors to read current literature such as:
- Proteome Res. 2014, 13, 6, 2761–2770
as well as read the cited, and cited by papers. As it stands, I do not understand the reasoning behind the selection of this particular protocol. - I find the instrument settings to be wholly insufficient for the purpose of phosphoproteomics. Not only the type of fragmentation was omitted (which is crucial), but the setting. Is the instrument in OTOT mode, or OTIT? HCD has been shown to be highly effective for phosphopeptides, however ETD is effective for higher charge state peptides. But was this CID? Or EThcD? There is no indication in the methods. Also, a decision tree strategy is highly advised with multiple injections, particularly given the instrument (Lumos) used for this analysis. For example the authors should be aware of:
- Am. Soc. Mass Spectrom. 2019, 30, 9, 1578–1585
Following this the MS2 scan range is completely strange. Phosphopeptides have a tendency to be multiply charged and so scan range of over 2000 can be needed. And why is the MS2 range start at 300 m/z??? And only go to 1250 m/z?? This should have been left at ‘auto’. Additionally an MS2 IT time of 20ms is far too low. 20ms is common for most peptides and not suitable for phosphopeptides. A minimum of 60ms (for ion trap) or 80ms for Orbitrap is required. Full instrument settings should be given in supplementary data. I feel that unless this is clarified, I cannot allow this to be published in this journal.
To be clear on this point: the MS strategy employed here is totally unsuitable for the workflow. It does not mean that the data is entirely wrong, but that the workflow used has seriously reduced the quantity of meaningful information that could have been obtained from these experiments. And therefore badly undermines the quality of this article. In light of the importance of the subject matter and the Impact Factor, I would insist that the proteomics workflow be re-run.
Indeed I must ask here whether the methods written are indeed the methods used. If there has been a mistake – for example a copy-and-paste methods from another paper, then I would insist that all raw files from the experiment be made available to the reviewers in order to check this for themselves. Indeed, submission of raw data to PRIDE should be performed in any case.
- Section 2.4 omits any meaningful description of how the phosphopeptide data was analysed. For example, are the authors aware of the R package PhosR? This allows for multiple processing pathways for this data, for example filtering and imputing, normalisation and analysis of signalling pathways etc.
Metascape needs to be cited. In fact this applies throughout the paper. Maxquant and Perseus should be cited. Please go through the entire article to ensure proper citations are included.
- Results and data analysis: there is a fundamental issue here with the selection of levels of significance, statistical tests and sample sizes. Whereas n=6 is considerably better than in most published proteomics experiments, the application of fold change (FC) > 1.3 (Log2(0.38)) is too low and could open the authors up to the accusation of ‘cherry picking’, particularly using t tests with no observable multiple test corrections such as Bonferroni. A minimum of FC > 1.5 at p < 0.05 should be used. With this data FC > 2 at p< 0.01 is strongly advised. Please correct this in the text as far as possible. It would be enlightening to be able to view the CVs for each quantification data point in order to evaluate (to some degree) what significance level may be appropriate.
Line 234 goes some way to help ameliorate the ‘cherry picking’ issue noted above by introducing an FC > 2, which is an improvement. However p values must be considered further. Actually are these p values, or adjusted p values? A curated output from Maxquant (and Perseus) should be included in the supplementary to assess these issues.
Figure 3B is the PCA plot and not the volcano plot. Concerning the PCA, it is clear that one replicate from HFD deviates along PC2. Given that PC1 is likely biological variation, PC2 is likely variance induced through measurement error. Have the authors attempted to re-evaluate the data with this replicate omitted? It may well improve p values and overall CVs. I strongly suggest the authors submit this data to a bioinformatician to assist them.
The sentence between lines 241 and 243 discusses the changes in protein levels as evidenced by the heatmap. And also discussed the two clusters. Firstly, what are these clusters? What unifies them or defines them? Indicate where lipid metabolism lies on this. Specifically what is it trying to express? In fact, given the shading, the figure is trying to imply a huge amount of differentially regulated proteins, which then begs the question ‘what units are the legend in’? Is that the FC range in base 10? Or base 2? The heatmap needs to be annotated considerably better.
Figure 4 is an improvement, and displays more insight.
The discussion of the phosphoproteins suffers in some ways due to the reasons mentioned above. Also I cannot tell if the authors are aware of the fundamental difference between the relative abundances of the phosphoproteins and the relative abundance of phosphorylation on those proteins, at the sites specified. This is done at the peptide level where it may be noticed that phosphorylation at a particular site changes, but the relative abundance of the protein itself does not change. This is represented as the relative abundance of a specifically site-phosphorylated peptide NOT protein abundance between experimental conditions. If this is simply an error in manuscript writing, then please correct this and make this very clear in the text. The supplementary data gives me no information on this.
Additionally a complete spread sheet containing the full proteomic analysis results should be part of the supplementary materials. Raw data and search engine results should be uploaded to a repository such as PRIDE, and access of this data to reviewers should be made available.

Author Response
Response to Reviewer 1 comments
Firstly I find the subject matter of high interest, and suitable for submission to a high impact journal. I consider the idea of combinatorial proteomics/lipidomics also of high value. In general I found the paper to be well written, clear and legible in good English with only a few spelling errors. However I found some significant issues that must be addressed. My speciality is in proteomics, so the authors will find no assessment of the lipidomics work within this review, and would request the Editor to send the paper to another appropriate reviewer.
At this particular time, I am advising that this article undergoes significant revisions, outlined below.
- I find the title needs to be clearer as the subject at hand is non-alcoholic fatty liver disease, not just fatty liver.
Response: Thank you for this comment. To make it clear, we have now revised the title as: “Multi-omics approach reveals dysregulation of protein phosphorylation correlated with lipid metabolism in mouse non-alcoholic fatty liver”.
- Line 75 of the introduction (“Our comprehensive analysis …”) to the end of the introduction should be removed. This should be in results or discussion. The introduction is no place for conclusions.
Response: Agreed, and removed as suggested. We rephrased and incorporated these elsewhere in discussion and conclusion.
- Section 2.2 needs further explanation: from a proteomics perspective the sample preparation protocol is slightly unusual. Whereas in-solution digestion is commonplace, more modern concepts such as S-Traps and SP3 are now widespread. This in itself is not an issue, however the use of iodoethanol in place of the more common IAA or CAA needs explanation, perhaps? A reference to a publication where this method is described fully is needed. As is the use of the barocycler. Please cite these properly. Sample preparation in phosphoproteomics is an active field and I strongly advise the authors to read current literature such as:
- Proteome Res. 2014, 13, 6, 2761–2770. as well as read the cited, and cited by papers. As it stands, I do not understand the reasoning behind the selection of this particular protocol.
Response: We have been using this method of sample preparation for quite some time, and results have been published extensively in dozens of publications by us and others. We have now added the original reference that describes this sample preparation protocol. This section has been further expanded, and additional references that used this method are also included.
- I find the instrument settings to be wholly insufficient for the purpose of phosphoproteomics. Not only the type of fragmentation was omitted (which is crucial), but the setting. Is the instrument in OTOT mode, or OTIT? HCD has been shown to be highly effective for phosphopeptides, however ETD is effective for higher charge state peptides. But was this CID? Or EThcD? There is no indication in the methods. Also, a decision tree strategy is highly advised with multiple injections, particularly given the instrument (Lumos) used for this analysis. For example the authors should be aware of:
Am. Soc. Mass Spectrom. 2019, 30, 9, 1578–1585
Following this the MS2 scan range is completely strange. Phosphopeptides have a tendency to be multiply charged and so scan range of over 2000 can be needed. And why is the MS2 range start at 300 m/z??? And only go to 1250 m/z?? This should have been left at ‘auto’. Additionally an MS2 IT time of 20ms is far too low. 20ms is common for most peptides and not suitable for phosphopeptides. A minimum of 60ms (for ion trap) or 80ms for Orbitrap is required. Full instrument settings should be given in supplementary data. I feel that unless this is clarified, I cannot allow this to be published in this journal.
To be clear on this point: the MS strategy employed here is totally unsuitable for the workflow. It does not mean that the data is entirely wrong, but that the workflow used has seriously reduced the quantity of meaningful information that could have been obtained from these experiments. And therefore badly undermines the quality of this article. In light of the importance of the subject matter and the Impact Factor, I would insist that the proteomics workflow be re-run.
Indeed, I must ask here whether the methods written are indeed the methods used. If there has been a mistake – for example a copy-and-paste method from another paper, then I would insist that all raw files from the experiment be made available to the reviewers in order to check this for themselves. Indeed, submission of raw data to PRIDE should be performed in any case.
Response: We profoundly regret for our negligence and appreciate the reviewer for bring this issue to our attention. Indeed, all the LC-MS data was collected in an Q-Exactive HF Orbitrap, not in an Orbitrap Fusion Lumos as we originally described. We apologize for this unthinkable mistake, and we have corrected and expanded the details of the settings used for the analysis. This is a great lesson for us and will make sure such mistaken does not happen again.
- Section 2.4 omits any meaningful description of how the phosphopeptide data was analysed. For example, are the authors aware of the R package PhosR? This allows for multiple processing pathways for this data, for example filtering and imputing, normalisation and analysis of signalling pathways etc. Metascape needs to be cited. In fact this applies throughout the paper. Maxquant and Perseus should be cited. Please go through the entire article to ensure proper citations are included.
Response: We apologize for missing detail description of phosphopeptide data analysis. The analysis of phosphopeptides, similarly to the global analysis, was performed using the Perseus platform. We are aware of the PhosR, and it has been used by our group in different projects but was not used for this particular study.
Results and data analysis: there is a fundamental issue here with the selection of levels of significance, statistical tests and sample sizes. Whereas n=6 is considerably better than in most published proteomics experiments, the application of fold change (FC) > 1.3 (Log2(0.38)) is too low and could open the authors up to the accusation of ‘cherry picking’, particularly using t tests with no observable multiple test corrections such as Bonferroni. A minimum of FC > 1.5 at p < 0.05 should be used. With this data FC > 2 at p< 0.01 is strongly advised. Please correct this in the text as far as possible. It would be enlightening to be able to view the CVs for each quantification data point in order to evaluate (to some degree) what significance level may be appropriate.
In addition to using 6 replicates for each condition, a minimum of 70% (5/6) replicates had a valid value (were quantified) in at least one group. We respectfully argue that a fold-change cutoff is an arbitrary cutoff value. The fold-change (and adjusted p-value) cutoff was decided before the sample collection to avoid biases, as it is standard practice. The cutoff was chosen based on previous literature reports that have used a FC cutoff of Log2(0.38)1-5. The values p-values used for the cutoff were FDR corrected p-values (q-values). We apologize for not clarifying it in the manuscript, and we have now made it explicit in the text.
Line 234 goes some way to help ameliorate the ‘cherry picking’ issue noted above by introducing an FC > 2, which is an improvement. However, p values must be considered further. Actually are these p values, or adjusted p values? A curated output from Maxquant (and Perseus) should be included in the supplementary to assess these issues.
Response: We have included the Perseus output data as a supplementary file. P-values used were FDR corrected values (q-values)
Figure 3B is the PCA plot and not the volcano plot. Concerning the PCA, it is clear that one replicate from HFD deviates along PC2. Given that PC1 is likely biological variation, PC2 is likely variance induced through measurement error. Have the authors attempted to re-evaluate the data with this replicate omitted? It may well improve p values and overall CVs. I strongly suggest the authors submit this data to a bioinformatician to assist them.
Response: We were aware of this outlier. Prior to original submission, we discussed this but did not remove/omit the replicate that deviates in the PCA plot as no obvious technical mistake was recorded or found. Neither any error in data entry/typos were found, nor the mice collected came from a different population. Further, no obvious difference in LC-MS raw data was also noticed. So, we could not justify the removal of such outlier.
However, as suggested by the reviewer, we indeed tested the data by removing the outlier sample. We identified 570 proteins with p-value ≤0.05 compared to the 613 proteins reported originally (without removing). Of those 570 significant proteins, 53 were new (not in the list of 613 proteins). Similarly, 100 proteins that were present in the list of 613, were not present after removing the outlier sample. Overall, 84% proteins were still identified as significant in both comparisons.
The sentence between lines 241 and 243 discusses the changes in protein levels as evidenced by the heatmap. And also discussed the two clusters. Firstly, what are these clusters? What unifies them or defines them? Indicate where lipid metabolism lies on this. Specifically, what is it trying to express? In fact, given the shading, the figure is trying to imply a huge amount of differentially regulated proteins, which then begs the question ‘what units are the legend in’? Is that the FC range in base 10? Or base 2? The heatmap needs to be annotated considerably better.
Response: The heatmap was clustered based on hierarchical clustering of protein z-scores. The color legend represents z-cored LFQ values. We have now described this in the figure legend. The impression that the heatmap is implying “huge amount of differentially regulated proteins” comes from the fact that the proteins here displayed are only the significant proteins. Thus, the difference between the two groups is expectedly lager than a total heatmap, and its intended to be a general representation of the “significantly regulated” population of our dataset.
Figure 4 is an improvement, and displays more insight.
Response: Thank you.
The discussion of the phosphoproteins suffers in some ways due to the reasons mentioned above. Also I cannot tell if the authors are aware of the fundamental difference between the relative abundances of the phosphoproteins and the relative abundance of phosphorylation on those proteins, at the sites specified. This is done at the peptide level where it may be noticed that phosphorylation at a particular site changes, but the relative abundance of the protein itself does not change. This is represented as the relative abundance of a specifically site-phosphorylated peptide NOT protein abundance between experimental conditions. If this is simply an error in manuscript writing, then please correct this and make this very clear in the text. The supplementary data gives me no information on this.
Response: We agree with the Reviewer’s comment, and yes, we are aware of the fundamental differences between the relative abundance of a phosphoprotein and the relative abundance of the phosphorylation on that protein. The phosphorylation data presented in this manuscript refers to the site specific phosphorylation changes, not the total phosphoprotein level changes. We indeed checked if increase or decrease in site-specific phosphorylation is simply due to changes in the protein level or it is directly related to site specific phosphorylation. It is important to note that the number of significant proteins were evenly distributed between up- and down-regulated proteins. However, majority of site-specific protein phosphorylation were down. This indicates that majority of changes in site-specific phosphorylation were not simply due to the changes in the protein level. We apologize if we failed to convey this clearly. We have tried to clarify this in the revised version. We also generated a scatter plot between protein intensity and site specific phosphorylation intensity of all significant proteins. If changes in the phosphorylation level is dependent on the changes in relative protein level, majority of the data should fall along the solid line (which represents an equal fold-change between phospho-site and global datasets), and this was not the case. This indicates that majority of phosphorylation level changes were also due to site-specific changes in protein phosphorylation. This confusion may have arisen from the wording used during manuscript writing. We have now made it clear when we were referring to site level changes. To clarify limitations, we have now added a line at the end to clarify this.
Additionally, a complete spread sheet containing the full proteomic analysis results should be part of the supplementary materials. Raw data and search engine results should be uploaded to a repository such as PRIDE, and access of this data to reviewers should be made available.
Response: Full results of proteomic analysis using Perseus are now provided in a spread sheet. All the raw and MaxQuant output files are also deposited in MassIVE (massive.ucsd.edu). The supplementary materials table S1 and S2 are now replaced with Perseus global and Perseus phospho analysis datasets.
1 Klähn, S. et al. Integrative analysis of the salt stress response in cyanobacteria. Biology Direct 16, doi:10.1186/s13062-021-00316-4 (2021).
2 Li, J., Paulo, J. A., Nusinow, D. P., Huttlin, E. L. & Gygi, S. P. Investigation of Proteomic and Phosphoproteomic Responses to Signaling Network Perturbations Reveals Functional Pathway Organizations in Yeast. Cell Reports 29, 2092-2104.e2094, doi:10.1016/j.celrep.2019.10.034 (2019).
3 Liu, R. et al. Proteomic Profiling Reveals Roles of Stress Response, Ca
2+
Transient Dysregulation, and Novel Signaling Pathways in Alcohol‐Induced Cardiotoxicity. Alcoholism: Clinical and Experimental Research 44, 2187-2199, doi:10.1111/acer.14471 (2020).
4 Singh, K. D. et al. Natural Genetic Variation Influences Protein Abundances in C. elegans Developmental Signalling Pathways. PLOS ONE 11, e0149418, doi:10.1371/journal.pone.0149418 (2016).
5 H, S. et al. Proteomic Studies on the Mechanism of Myostatin Regulating Cattle Skeletal Muscle Development. Front Genet 12, doi:10.3389/fgene.2021.752129 (2021).

Reviewer 2 Report
In the Kim et al work, the authors have pursued a multi-omics approach to clarify the altered processes in HFD versus LFD mice livers. In my opinion, the work is interesting to read and opens the way for further investigations. Nevertheless, some comments are addressed to the authors.
- Did the authors perform technical replicates for the mass spectrometry measurements? If yes, how the authors have treated them?
- How many micrograms of protein extracts have been injected into the instrument for the proteomics analysis? It should be specified in the method section.
- In the “Data analysis” paragraph of the Methods section, the authors specified the number of amino acids minimum required for, I suppose, peptide identification (line 144). Did they use a peptide number cut off for protein identification? Then, the protein identification “quality parameters” (e.g. number of peptides assigned to each protein or % of sequence coverage) have not been reported in Supplementary Table S1. It should be added as additional information.
- 1A is not cited in the text and, in the caption, the meaning of asterisk is not specified, as I suppose, p<0.05.
- In line 226 the authors referred to Fig.3A as a PCA plot, but it is not.
- In Fig. 3A, I did not fully understand if the authors reported in the y-axis the -log10 of q values as cited in the main text (line 230). If it would be the case, how did they treat the zero values reported in Supplementary Table S1? In addition, if Fig.3C is an insert of Fig.3A, why do they have not the same y-axis scale? Inspecting Supplementary Table S1, I think that you reported p-values for the y-axis of Fig.3C. If you plotted q values in Fig. 3A, you should report the same values also in 3C. Finally, to give more evidence to the most dysregulated proteins in Fig. 3C, I would suggest making the x-axis a little more populated because, at a first glance, not all the proteins seem to have |log2FC|>2.
- In Fig.4A the GO Biological Processes terms are not very clearly readable. In addition, in Fig. 4B I did not understand where the arrow startings are pointed. Do they indicate the leading term of a group? For example, considering the cluster “small molecule catabolic process”, the arrow is pointed towards an edge between two nodes: one all red-colored (and specific for the up-regulated proteins) and one represented by about 75% of up- and 25% of down-regulated proteins. Considering the heatmap in Fig.4A, I think the arrow refers to the up and down shared node, but also the other processes are not clearly indicated by arrows in Fig. 4B.
- Line 338: did the authors intend Fig.5C instead of Fig.5B?
- In lines 472-473 the authors wrote: “While we did not identify Pparγ, we identified up-reguation of Pparα”. Did you mean that you identified Pparα up-regulated in your dataset? I can’t find it in Supplementary Table S1.
- Could the authors explain how they obtained the heatmaps in Fig.7? Based on the VIP score reported in Supplementary Table S5, Lgals1 is the top-ranked. Why is it not present in the heatmap in Fig.7A? Moreover, considering that it is also the best-ranked in terms of p-value (and one of the most dysregulated) in the global proteomics analysis according to Fig.3C, it might deserve a comment maybe in the discussion section. Indeed, inspecting the recent literature Lgals1 has been demonstrated to be involved in obesity (PMID: 33431823) and its targeting would be a possible therapeutic strategy (PMID: 26880535, PMID: 25920776, PMID: 28444942).
- The Supplementary material section description does not agree with the loaded Tables.
Author Response
Response to Reviewer 2 comments
In the Kim et al work, the authors have pursued a multi-omics approach to clarify the altered processes in HFD versus LFD mice livers. In my opinion, the work is interesting to read and opens the way for further investigations. Nevertheless, some comments are addressed to the authors.
Response: We thank the reviewer for providing insightful evaluation of our manuscript as well as helpful suggestions.
- Did the authors perform technical replicates for the mass spectrometry measurements? If yes, how the authors have treated them?
Response: We thank the reviewer for raising this point. We don’t run technical replicates of the samples but we run Hela digest to check instrument performance and instrument reproducibility. This is a routine practice in our lab, and we do this after each calibration. Calibration is done in every 72 hours or earlier if we see a mass shirt >5 ppm. LC-MS reproducibility is monitored regularly by calculating CV (coefficient of variation) of both peptides and proteins. Our CV is always around 10% for peptides and around 5% for protein LFQ intensity. In our biological runs, >82% of the quantified proteins were present in 5 out of 6 liver samples, clearly showing high reproducibility of the data.
- How many micrograms of protein extracts have been injected into the instrument for the proteomics analysis? It should be specified in the method section.
Response: We apologize for not indicating this information. The following information has now been included in the Materials and Methods section
“An aliquot of each sample containing 500 ug of total protein was precipitated using ice-cold acetone at -20 °C overnight. After acetone removal, protein pellets were reduced with 10mM dithiothreitol in 8M urea, and alkylated using iodoethanol in ACN (2% iodoethanol, 0.5% triethylphosphine, 97.5% acetonitile). Proteins were digested with mass spec grade Trypsin/LysC mix (Promega) at a 1:50 (w/w) enzyme-to-substrate ratio, using a barocycler (Pressure BioScience Inc.) at 50°C with 60 cycles of 20kpsi for 50 seconds and 1 atmospheric pressure (1 ATM) for 10 seconds. Samples were cleaned using Pierce Peptide Desalting Spin Columns (Thermo Fisher Scientific). Samples were then divided into two tubes containing approximately 50μg and 450μg peptides. The 450μg peptides were further processed for phosphopeptide enrichment using PolyMac phosphopeptide enrichment spin tips (Tymora Analytical) using manufacturer’s recommendataion. Finally, samples were dried and samples for global proteomics were dissolved in 20μl of 0.1% FA in 3% ACN and 1 μl was loaded to the LC column. Purified phosphopeptides were re-suspended in 10μl of 0.1% FA in 3% ACN and 5 μl was used phosphoproteomics analysis.
- In the “Data analysis” paragraph of the Methods section, the authors specified the number of amino acids minimum required for, I suppose, peptide identification (line 144). Did they use a peptide number cut off for protein identification? Then, the protein identification “quality parameters” (e.g. number of peptides assigned to each protein or % of sequence coverage) have not been reported in Supplementary Table S1. It should be added as additional information.
Response: We apologize for not indicating this information. We used at least 1 unique peptide and at least 2 MS/MS counts per protein for protein identification. WE have now included this information in the revised manuscript.
- 1A is not cited in the text and, in the caption, the meaning of asterisk is not specified, as I suppose, p<0.05.
Response: We apologize for not indicating this information. Figure 1A is now described in the text and figure legend now explains what asterisk means.
- In line 226 the authors referred to Fig.3A as a PCA plot, but it is not.
Response: We apologize for the confusion. Figure 3A and Figure 3B are now switched to match descriptions in the text and figure legend is updated accordingly.
- In Fig. 3A, I did not fully understand if the authors reported in the y-axis the -log10of q values as cited in the main text (line 230). If it would be the case, how did they treat the zero values reported in Supplementary Table S1? In addition, if Fig.3C is an insert of Fig.3A, why do they have not the same y-axis scale? Inspecting Supplementary Table S1, I think that you reported p-values for the y-axis of Fig.3C. If you plotted q values in Fig. 3A, you should report the same values also in 3C. Finally, to give more evidence to the most dysregulated proteins in Fig. 3C, I would suggest making the x-axis a little more populated because, at a first glance, not all the proteins seem to have |log2FC|>2.
Response: We thank the reviewer for this useful comment. To clarify, we used the -log10 of q values in determining significantly differently expressed proteins and accordingly, we changed y-axis unit in Fig 3A and 3B into -log10 of q values. Zero q values were converted to: (10^-1) * (current lowest non-zero q-value).
Yes, we agree with the reviewer that x-axis of the Fig 3C needs to show more number marking. Fig 3C now shows -8, -6, -4, -2, 0, 2, 4, 6, 8 on x-axis.
- In Fig.4A the GO Biological Processes terms are not very clearly readable. In addition, in Fig. 4B I did not understand where the arrow startings are pointed. Do they indicate the leading term of a group? For example, considering the cluster “small molecule catabolic process”, the arrow is pointed towards an edge between two nodes: one all red-colored (and specific for the up-regulated proteins) and one represented by about 75% of up- and 25% of down-regulated proteins. Considering the heatmap in Fig.4A, I think the arrow refers to the up and down shared node, but also the other processes are not clearly indicated by arrows in Fig. 4B.
Response: We thank the reviewer for raising this issue. Figure 4A is now enlarged to be easily readable. In addition, Figure 4B now displays dotted line to clearly show distinct GO Biological Process and nodes that belong to each GO term.
- Line 338: did the authors intend Fig.5C instead of Fig.5B?
Response: We apologize for the confusion. Yes, we corrected the text.
- In lines 472-473 the authors wrote: “While we did not identify Pparγ, we identified up-reguation of Pparα”. Did you mean that you identified Pparα up-regulated in your dataset? I can’t find it in Supplementary Table S1.
Response: We apologize for the confusion. Yes, the reviewer’s point was right, and we changed the text.
- Could the authors explain how they obtained the heatmaps in Fig.7? Based on the VIP score reported in Supplementary Table S5, Lgals1 is the top-ranked. Why is it not present in the heatmap in Fig.7A? Moreover, considering that it is also the best-ranked in terms of p-value (and one of the most dysregulated) in the global proteomics analysis according to Fig.3C, it might deserve a comment maybe in the discussion section. Indeed, inspecting the recent literature Lgals1 has been demonstrated to be involved in obesity (PMID: 33431823) and its targeting would be a possible therapeutic strategy (PMID: 26880535, PMID: 25920776, PMID: 28444942).
Response: We thank the reviewer for the comment. The heatmaps in figure 7 are autogenerated on the statistical package JMP from SAS. The software does not allow to display all the proteins/phosphosites as row titles and therefore showing only a subset in the list. For this reason, we provided the entire heatmap results in Supplementary Material TableS5 & S6. Second, we agree that upregulation of Lgals1 in DIO animals is worth noting, and thus we highlighted its role in steatosis in discussion section.
- The Supplementary material section description does not agree with the loaded Tables.
Response: We apologize for this discrepancy, and we have now addressed this issue.

Reviewer 3 Report
Comments on Manuscript ID: cells-1620011
The present manuscript by Kim et al. is generally well written and the presentation of the processed data is good. The experimental approaches are, a priori, robust. The global proteomics and phosphoproteomics raw data have good quality, showing nice Total Ion Chromatograms along the LC gradient and similar intensity on the different injections. Peptide elution profiles are generally good as well.
However I have some minor concerns that should be clarify by the authors.
Line 115: The manufacterer digestion protocol for Trypsin/LysC mix is quite different to the approach used in this study, with a single digestion temperature (50oC) at high-pressure. If this approach has been previously used please include reference. If not the authors should explain why the standard protocol was not used.
Line 122: “Samples were analyzed by reverse-phase LC-ESI-MS/MS system using the Dionex UltiMate 3000 RSLC nano System coupled to the Orbitrap Fusion Lumos Mass Spectrometer”. However, according to the dataset webpage at MassIVE repository an Orbitrap QExactive HF was used. There is an urgent need to clarify this.
Line 156: “Significantly up-or-downregulated proteins between the two groups 156 were determined by Student’s t-test with Permutation-based FDR (q-value < 0.05, log(FC) 157 > 0.38)”. Later, in the volcano plots figures, the vertical axis is labeled with –Log10P. What is the meaning of this P? If the authors have use a threshold of q-val ≤0.05 to select modulated proteins, why not Log10q-val?
In the link that the authors provided to the Editor the number of raw files for the phosphoproteomic analysis is only three in the case of LFD samples. Why not six? If only three LFD samples have been analyzed this discrepancy should be explained in the text. Also, please discuss how this could compromise the phosphoproteomic analysis.
A targeted proteomic analysis of selected candidates by, for instance, Parallel Reaction Monitoring in the same Orbitrap platform, would give an extra strenght to the analysis. Would be nice to discuss this and other limitations of the study in the proper section of the manuscript.
Author Response
Response to Reviewer 3 comments
The present manuscript by Kim et al. is generally well written and the presentation of the processed data is good. The experimental approaches are, a priori, robust. The global proteomics and phosphoproteomics raw data have good quality, showing nice Total Ion Chromatograms along the LC gradient and similar intensity on the different injections. Peptide elution profiles are generally good as well.
Response: We thank the reviewer for taking the time to go over our raw files and observing their good quality.
However, I have some minor concerns that should be clarify by the authors.
Line 115: The manufacterer digestion protocol for Trypsin/LysC mix is quite different to the approach used in this study, with a single digestion temperature (50oC) at high-pressure. If this approach has been previously used please include reference. If not the authors should explain why the standard protocol was not used.
Response: We thank the reviewer for bringing this to our attention. The Core has a published reference protocol that has been used in many studies and is robust although we recognized is outdated and we are working on up-dating it. The following citation has been included in the text. Hedrick VE, LaLand MN, Nakayasu ES, Paul LN. Digestion, Purification, and Enrichment of Protein Samples for Mass Spectrometry. Curr Protoc Chem Biol. 2015 Sep 1;7(3):201-222. doi: 10.1002/9780470559277.ch140272. PMID: 26331527. We have extensively published proteomic results that utilized this digestion protocol, and a few latest references are also included in the text.
Line 122: “Samples were analyzed by reverse-phase LC-ESI-MS/MS system using the Dionex UltiMate 3000 RSLC nano System coupled to the Orbitrap Fusion Lumos Mass Spectrometer”. However, according to the dataset webpage at MassIVE repository an Orbitrap QExactive HF was used. There is an urgent need to clarify this.
Response: We greatly appreciate the reviewer for bringing this to our attention and avoiding further confusion. Indeed, the data was collected in an Q-Exactive Orbitrap HF and not on an Orbitrap Fusion Lumos. We profoundly apologize for this unthinkable mistake, and we have corrected and expanded the details of the settings used for the analysis. This has been a great lesion and will make sure such mistaken does not happen in the future.
Line 156: “Significantly up-or-downregulated proteins between the two groups 156 were determined by Student’s t-test with Permutation-based FDR (q-value < 0.05, log(FC) 157 > 0.38)”. Later, in the volcano plots figures, the vertical axis is labeled with –Log10P. What is the meaning of this P? If the authors have use a threshold of q-val ≤0.05 to select modulated proteins, why not Log10q-val?
Response: The P value in the y-axis referred to the adjusted P-value (q-value). We apologize for any confusions this has caused. We have changed the labeling to make it clearer.
In the link that the authors provided to the Editor the number of raw files for the phosphoproteomic analysis is only three in the case of LFD samples. Why not six? If only three LFD samples have been analyzed this discrepancy should be explained in the text. Also, please discuss how this could compromise the phosphoproteomic analysis.
Response: Six animals were used for each LFD and HFD and for both global and phospho analyses. There were a total of 12 datasets for global and 12 data sets for phosphoproteomics. We indeed provided link with all 12 samples for each group, and not sure why only 3 LFD samples were present. We apologize if this was a technical problem from our end, and thank the reviewer for point this out. We have carefully revised the folder shared with the reviewer for any incomplete upload since the global and phosphoproteomics were performed in the same number of samples precisely not to compromise the analysis. In MassIVE there are 24 LC-MS raw files comprising 12 for global and 12 for phosphoproteomics ftp://massive.ucsd.edu/MSV000088835/.
A targeted proteomic analysis of selected candidates by, for instance, Parallel Reaction Monitoring in the same Orbitrap platform, would give an extra strenght to the analysis. Would be nice to discuss this and other limitations of the study in the proper section of the manuscript.
Response: We have included this limitation in the conclusion part of the revised manuscript, and would be part of our future study.

Round 2
Reviewer 1 Report
MS raw files must be uploaded to PRIDE and an accession number included in the paper.